# Chromatin and Cancer: Implications of Disrupted Chromatin Organization in Tumorigenesis and Its Diversification

**DOI:** 10.3390/cancers15020466

**Published:** 2023-01-11

**Authors:** Poonam Sehgal, Pankaj Chaturvedi

**Affiliations:** Department of Cell and Developmental Biology, University of Illinois at Urbana-Champaign, Urbana, IL 61801, USA

**Keywords:** cancer, chromatin, epigenetic, gene expression, nucleus, therapeutics

## Abstract

**Simple Summary:**

Human DNA is ~2 m long, but is efficiently packaged in cell nucleus that is a million times smaller. The concerted action of several enzymes enables this packaging through hierarchical folding of DNA. These enzymes deposit unique marks on the DNA and DNA-binding proteins collectively called the histone code. The histone code determines when and how a segment of DNA will open and be expressed. Pathogenic changes or mutations in the proteins produced by the cells alter this expression pattern. Here, we discuss how these mutations affect DNA packaging in cancer establishment, diversification, and therapeutic resistance. We also document available therapeutic approaches aimed at DNA packaging in cancer and the direction in which current research is heading.

**Abstract:**

A hallmark of cancers is uncontrolled cell proliferation, frequently associated with an underlying imbalance in gene expression. This transcriptional dysregulation observed in cancers is multifaceted and involves chromosomal rearrangements, chimeric transcription factors, or altered epigenetic marks. Traditionally, chromatin dysregulation in cancers has been considered a downstream effect of driver mutations. However, here we present a broader perspective on the alteration of chromatin organization in the establishment, diversification, and therapeutic resistance of cancers. We hypothesize that the chromatin organization controls the accessibility of the transcriptional machinery to regulate gene expression in cancerous cells and preserves the structural integrity of the nucleus by regulating nuclear volume. Disruption of this large-scale chromatin in proliferating cancerous cells in conventional chemotherapies induces DNA damage and provides a positive feedback loop for chromatin rearrangements and tumor diversification. Consequently, the surviving cells from these chemotherapies become tolerant to higher doses of the therapeutic reagents, which are significantly toxic to normal cells. Furthermore, the disorganization of chromatin induced by these therapies accentuates nuclear fragility, thereby increasing the invasive potential of these tumors. Therefore, we believe that understanding the changes in chromatin organization in cancerous cells is expected to deliver more effective pharmacological interventions with minimal effects on non-cancerous cells.

## 1. Introduction

In metazoans, spatiotemporally coordinated gene expression is essential for cellular proliferation and fate determination. Disruption of controlled gene expression can have deleterious, and frequently pathogenic consequences. The development of cancer is a classic example where cell fate determination, proliferation, cell–cell interaction, or signaling are adversely impacted by deregulated gene expression. It is, therefore, not surprising that tumors with subtypes of more than 100 origins have been identified [1]. Regardless of their origins and the involved pathways, all cancers share certain hallmarks such as sustained proliferation, evasion of apoptosis, insensitivity to growth suppressors, self-sufficiency in expressing growth factors, invasive metastatic potential, and sustained angiogenesis [1]. A common factor among these hallmarks of cancer is genomic instability and structural disorganization (Figure 1).

The gene expression is largely controlled by chromatin, an ensemble of DNA, histones, and several non-histone proteins [11]. The hierarchical chromatin organization ensures the necessary compaction to accommodate entire cellular DNA in the nucleus. It also regulates access to transcriptional, repair, or replicative factors ensuring that informational continuity is maintained despite the damage induced by the environment. The link between oncogenesis and transcriptional dysregulation is multifaceted and involves chromosomal rearrangements, chimeric transcription factors, or altered epigenetic marks. The epigenetic marks on cells derived from the same tumor differ both at local and global levels, suggesting that the altered epigenetic marks can be oncogenic drivers or may lead to tumor cell heterogeneity and therapeutic resistance. Notably, genome-wide mapping of diverse tumors has established that almost half of the driver mutations are chromatin regulators [12]. An ample amount of literature on oncogenesis is devoted to the role of individual chromatin modifiers, transcription factors, and novel fusion proteins [1,13,14,15]. Therefore, the current review presents a unique perspective on how disruption of higher-order chromatin and the associated chromatin remodelers contribute to cancer etiology. A fundamental understanding of chromatin unraveling in neoplastic transformation and diversification can play a vital role in designing preventive or therapeutic approaches for cancer patients.

## 2. Chromatin: The Fundamental Unit of Nuclear Functionality

Chromosomal DNA is hierarchically packaged into chromatin, ensuring functionality and integrity. The most basic layer of this packaging is regularly spaced coiling of ~146 bp of DNA on nucleosomes, an octameric scaffold made of two copies each of core histones (H2A, H2B, H3, and H4) [11,16] (Figure 2a). The compaction is further enabled by the binding of linker histone H1 at the DNA entry/exit sites on these nucleosomes [11]. The sequence of these histone proteins is evolutionarily conserved, and the net positive charge of the histones neutralizes the negative charge on DNA allowing it to be compactly packaged within the confines of the nucleus. Despite the cationic charge on histones, the net charge on the 10 nm fiber is still negative and is further neutralized by several non-histone proteins, cations, and the local electrostatic environment during the higher-order chromatin compaction. Modulation of this electrostatic interaction, either by ionic balance at the local level or by the chromatin remodeling complexes, forms the fundamental premise of chromatin structure and function [17]. Interestingly, negative charge on RNA also antagonizes the charge balance on the nucleosomes and leads to chromatin decompaction [18].

The intrinsically disordered amino-terminal histone tails protruding out of the nucleosomes provide an additional layer of regulation of chromatin function. The “histone code” of the post-translational modifications (PTMs) such as acetylation, methylation, phosphorylation, sumoylation, and ubiquitinylation on these histone tails regulate the binding of different chromatin modulators. The histone code, therefore, determines the fate of local chromatin compaction [19] (Figure 2b). Over the past two decades, a plethora of histone mark readers, writers, and erasers have been identified which are responsible for chromatin regulation. In a very simplistic view, hypoacetylation of chromatin causes compaction and is the hallmark of repressive chromatin, while hyperacetylation counters this state by weakening the internucleosomal interactions and promoting chromatin opening.

## 3. Chromatin Organization and Function during Interphase

The interphase chromatin observed almost a century ago by Emil Heitz as differentially stained euchromatin and heterochromatin [20] is the quintessential unit of large-scale genome organization and function. The advent of genome-wide mapping technologies has revealed that this segregation into euchromatin and heterochromatin is non-random and is of paramount importance for cell survival. Euchromatin represents an actively transcribed, early replicating fraction of the genome, while heterochromatin is a transcriptionally inactive, repeat-rich, late-replicating environment. Based on the state of compaction throughout the cell cycle, the heterochromatin is subdivided into facultative or constitutive heterochromatin. The facultative heterochromatin is a transcriptionally silent region of chromosomes, which can open and provide spatiotemporally regulated expression of tissue-specific genes [21]. The constitutive heterochromatin found at centromeric and pericentromeric regions remains compacted and is highly enriched in transposable elements and repeat sequences (Figure 1). The constitutive heterochromatin is hypothesized to reinforce the structural integrity of the nucleus [11], ensuring faithful chromosomal segregation during mitosis, and serving as an “evolutionary laboratory” for novel genes [22]. Curiously, even the paradigm of heterochromatin as a repressive environment is changing and studies have identified different heterochromatin subtypes, each with a unique set of epigenetic marks and functions [23].

Further, the positioning of the chromatin within the nucleus is deterministic, where expressed genes are positioned towards nuclear speckles to promote transcription, while the repressed genome is oriented away from these bodies [24,25]. Conversely, the gene-poor heterochromatin anchors to nuclear lamina [26] or nucleolus [27] through membrane-associated proteins such as lamins [28], and Lamin B-receptor (LBR) [29]. These heterochromatic segments at the nuclear periphery, referred to as Lamina Associated Domains (LADs), together comprise ~40% of the total genome, and can vary both in size (tens of megabases) and number (up to 1400) [30]. Summarily, the compacted chromatin in eukaryotic nuclei serves three basic but inter-related functions: regulating gene expression by controlled accessibility of transcriptional or replication machinery, providing rigidity to the nucleus to resist deformations, and protecting genomic material from chemical or radiation damages [11]. How these local changes alter chromatin organization and function in human diseases will be subsequently discussed in detail.

## 4. Pathological Consequences of Chromatin Abnormalities

The traditional view of cancer development is centered around the derepression of certain oncogenes or loss of tumor-suppressor genes, either caused by driver mutations or by the global disruption of epigenetic marks. However, a larger and often undetected effect on gene expression is caused by heterochromatin disorganization, and this phenomenon needs to be viewed holistically as an oncogenic driver. This disruption in heterochromatin can facilitate the initiation of tumorigenesis, the diversification of cells within the tumor, or a combination of both. This hypothesis is substantiated in several inherited disorders where the unraveling of chromatin occurs [28,31,32]. A few such examples are laminopathies, premature aging, Lynch syndrome, and thalassemia (Table 1). A notable aspect of these pathologies is the increased susceptibility to cancer, suggesting a commonality in the progression of the two categories of diseases [33,34,35]. In Table 1 we present a few examples of such chromatin modifiers, how they affect the nuclear organization in pathologies, and the associated risk of cancer development. These common features of inherited disorders and cancer can provide more effective therapeutic interventions.

### 4.1. Reactivation of X-Chromosome (XaXa) and Cancer

The most dramatic example of loss of heterochromatin in cancer progression is the loss of Barr bodies in certain breast and ovarian cancers [13,53,54]. The Barr body is an inactivated and highly condensed X-chromosome in female somatic cells, frequently found associated with the nuclear periphery. A low or absent Barr body count has been correlated with the high invasive potential of tumor cells and poor prognosis of patients [54]. About 50-fold increased risk of breast cancer has also been documented in males diagnosed with Klinefelter syndrome (XXY genotype) [55], providing vital clues to X chromosome dosage in cancer. Indeed, overexpression of several genes regulating chromatin organization or transcriptional regulation located on X-chromosomes has been implicated in cancers [56] (Figure 3). Additionally, mutations in the lysine demethylases found on X chromosomes (*KDM*6A and *KDM*5C) or their Y chromosome paralogs (*UTY* and *KDM*5D) have been linked to increased cancer risk, poor prognosis, or insensitivity to certain therapies [57]. However, it may be an indirect effect of their genome-wide demethylase activity leading to the derepression of proto-oncogenes. So far, the most pragmatic explanation for the XaXa genotype is that the inactive X chromosome (Xi) is spontaneously lost, and subsequent mitotic segregation errors cause duplication of the active X chromosome (Xa). However, similar outcomes can also be attributed to the loss of heterochromatin-associated epigenetic marks and factors or a partial aneuploidy.

### 4.2. Chromosomal Rearrangements Induced by Repetitive Elements

The presence of repetitive sequences and transposable elements in the genome is pivotal for the evolution of novel gene families, ncRNAs, and regulatory factors [22]. However, this dynamic component of the mammalian genome can also induce cis- or trans-chromosomal rearrangements through mechanistic breakage or recombination, occasionally leading to pathological outcomes. A classic example of this phenomenon is *GSTM*1 allelic polymorphism [60]. This gene belongs to a larger superfamily of glutathione transferase (GST) metabolic enzymes and has been found to have spontaneous deletions in ~50% of the human population through homologous recombination between repeats [60,61]. The glutathione transferases provide a protective role from oxidative stress by reactive oxygen species and by detoxification of xenobiotic molecules. Therefore, polymorphism in this gene family can influence the efficacy of different therapeutic agents and increase an individual’s susceptibility to certain toxins or carcinogens. Indeed, polymorphism of *GSTM*1 has been associated with an increased risk of breast and bladder cancers [62]. Similar pathological outcomes by rearrangements in flanking repetitive DNA have been observed for genes encoding Apolipoprotein B mRNA editing enzyme, catalytic polypeptide-like (*APOBEC*3B), UDP-glucuronosyltransferase (*UGT2B*17), and heptaglobins [61].

Fragile sites are the hotspots for chromosomal rearrangements induced by certain replication stresses and have long been studied relative to several diseases, including cancer. The proximity of certain constitutive fragile sites to protooncogenes was earlier proposed as a diagnostic marker for tumors [63] and speculations about their relationship to neoplasia were made. Such predictions were verified almost a decade later when the location of a fragile site (*FRA*11B) within trinucleotide repeats (CCG) at the 5′ end of a proto-oncogene *CBL*2 was mapped in Jacobsen syndrome (11q^−^) patients [64].

### 4.3. Altered Long-Range Chromatin Contacts in Cancer

The juxtaposition of unrelated super-enhancers or other cis-regulatory factors during chromosomal rearrangement may upregulate the expression of silenced or minimally expressed genes. A prime example of such regulatory rearrangement of a super-enhancer is at the *MYB* locus, first reported in adenoid cystic carcinoma (ACC) [65]. Here, the gain of the super-enhancer leads to the elevated expression of MYB transcription factor, which is further reinforced through a positive feedback loop. Similarly, the gain of enhancers or “enhancer-hijacking” has also been implicated in lineage-ambiguous leukemia, where the aberrant expression of a zinc-finger transcription factor essential for T-cell development (*BCL11*B) has been reported [66].

Interestingly, two distinct mechanisms of the gain of active enhancers at the *BCL*11B locus have been identified [66]. The first observation was chromosomal rearrangements that position the hematopoietic and stem progenitor cell (HSPC) super-enhancer within a few hundred kilobases of the *BCL*11B gene, leading to its ectopic expression in HSPCs. Another mechanism was the de novo formation of a super-enhancer by tandem amplification of a 2.5 kb noncoding sequence ~750 kb downstream of the *BCL*11B gene. The amplified sequence was found to have enhanced H3K27ac marks and formed multiple long-range chromatin loops within the *BCL*11B gene, indicative of its potent enhancer activity [66]. Such focal amplification of genomic segments containing super-enhancers has also been observed in other cases. In head and neck squamous and endometrial carcinomas, ~110–160 kb-sized focal amplifications of super-enhancers (KLF-HNSE) on chromosome 13q arm were observed to upregulate expression of Krüppel-like transcription factor-5 (KLF5) [67]. The authors also found amplified super-enhancers driving expression of oncogenes *MYC*, *USP*12, *PARD*6B, and *KLF*5 in uterine corpus endometrial carcinoma, colorectal carcinoma, liver hepatocellular carcinoma, and esophageal carcinomas, respectively [67].

In addition to tissue-specific gains, spontaneous deletions of super-enhancers can also have similar effects. Reduced expression of a splice variant of Regulator of calcineurin 1 (*RCAN*1.4) caused by the deletion of such a regulatory super-enhancer has been implicated in increased breast cancer cell metastasis [68]. Therefore, it appears that the oncogenic transformation through an altered regulatory potential of cis-regulatory elements, frequently located hundreds of kilobases away, is as pathogenic as mutations in the coding regions of genes. It is of no surprise that the oncogenic human viruses (e.g., Epstein–Barr Virus, Human Papilloma Virus) have evolved to hijack this mechanism of nucleating super-enhancers for sustained transcription of viral oncogenes [69]. Overexpressed viral and host proteins confer a selective growth advantage to the infected cells and can facilitate cancer development in the immunocompromised hosts.

Topologically associating domains (TADs) are another distal regulatory feature reported to be disrupted during neoplastic transformations. A study comparing normal and cancerous prostate cells found that the genome reorganization led to the formation of ~1000 cancer-specific TADs [70]. The average size of these cancer-specific TADs was significantly smaller than the normal TADs, and it shielded repression of prostate cancer-related genes such as *CBX2*, *CBX4*, *CBX8*, *TBX3,* and *PRMT6* [70]. Large-scale reorganization of the genome has also been seen in the case of normal B-cell development and leukemia [71], and cervical cancer where ~24% of A/B compartments were altered [72]. Loss of TADs was more drastic (~64%) in the case of triple-negative breast cancer (TNBC) compared to normal mammary epithelial cells [73]. Reorganization of TADs and insulator boundaries is a normal developmental process, but cancerous cells establish tumor-specific 3D genome architecture to facilitate oncogenic transcription.

### 4.4. Altered DNA Methylation in Cancer

DNA methylation of CpG dinucleotides is the process where methyl residue is covalently linked to the 5th position on cytosine residues by de novo (*DNMT*3A and *DNMT*3B) or maintenance (*DNMT*1) DNA methyl transferases [74]. Barring the CpG sequences upstream of the transcriptionally active sites, this epigenetic modification is found throughout the genome. The CpG methylation ensures the silencing of transposons and retroviral elements by limiting access to transcription factors in the compacted heterochromatin [74]. This phenomenon is evident in tumors where the loss of CpG methylation leads to an unwarranted transcription in pericentromeric heterochromatin of chromosomes 1 and 16, facilitating chromosomal instability and mitotic recombination events [75]. Transcription of hypomethylated satellite 2 DNA is frequently observed in the ICF syndrome, and tumors (e.g., Wilms, breast, and ovarian tumors). A direct correlation between the levels of satellite 2 transcripts and metastatic potential has been established for multiple carcinomas [76].

The aberrant transcription of hypomethylated repetitive D4Z4 repeats has also been identified as a causal factor of FSHD (Facioscapulohumeral muscular dystrophy). In this case, the hypomethylated D4Z4 repeats facilitate transcription of lncRNA DBE (D4Z4 binding element), which in turn promotes transcription of the *DUX*4 gene [77]. Another such example is the expression of lncRNA TNBL (Tumor-associated NBL2 transcript) in colorectal cancer, which is expressed from the hypomethylated NBL2 repeats located on acrocentric chromosomes 13, 14, 15, and 21 [78]. These TNBL transcripts were found to localize predominantly in the perinucleolar heterochromatin, suggesting its role in chromatin organization. The hypomethylation of repetitive sequences is a common feature among all known major cancer types [77]. However, this feature is not exclusive to cancers and is also seen in the pathologies associated with senescence or aging, such as atherosclerosis, cardiovascular diseases, and rheumatoid arthritis [77,79].

In contrast to global hypomethylation, DNA hypermethylation in cancers is predominantly localized to the promoters of expressed genes [80]. However, the field is divided over the notion of tumor-specificity of promoter hypermethylation. In a recent comparative meta-analysis of 18 distinct cancers and 22 different genes, Bouras et al. found that the promoter hypermethylation of several genes, and not any particular gene, was associated with a specific cancer type [81]. For instance, bladder cancer patients had elevated promoter methylation of *RASSF*1, *CDH*1, *DAPK*, and *CDKN*2A genes, which was distinct from hypermethylated genes *MGMT*, *FHIT*, and *hMLH*1 found in NSCLC (Non-Small Cell Lung Cancer) patients [81]. Overall, the promoter hypermethylation leads to the silencing of genes involved in pathways such as apoptosis (e.g., *DAPK*), cell cycle regulation (e.g., *CDKN*2A), cellular signaling (e.g., *APC*), DNA damage repair (e.g., *MGMT*), cell adhesion (e.g., *CDH*1), and detoxification (e.g., *GSTP*1) [82,83], thereby providing a growth advantage to the tumor cells.

Interestingly, non-CG methylation has also been observed on the gene bodies of highly expressed genes in the pluripotent cells [84]. The non-CG methylation on gene bodies observed in the stem cells but not in differentiated cells warrants investigations as to if similar mechanisms are responsible for the escape of silenced oncogenes. Hypermethylated gene bodies have recently been identified as a novel mechanism of induced gene expression of homeobox genes in several tumors [85]. As both hyper- and hypo- methylation has been implicated in the pathogenesis of cancer, the non-specific global demethylation induced by DNA methylation inhibitors can have a potential side-effect on the expression of non-CG methylated genes.

### 4.5. Chromatin and Alternative-Splicing of Pre-mRNA

In higher eukaryotes, alternative splicing or exon-skipping of pre-mRNA is important for the expression of multiple protein isoforms and the diversification of protein repertoires. However, recent studies have identified abnormal splicing in cancer and other hereditary diseases [86,87]. Studies have established chromatin and its modifiers as one of the key regulators that govern the splicing and inclusion of alternative exons in pre-mRNAs. Association of chromatin remodeler SWI/SNF (mating-type switch/sucrose nonfermenting) complex protein Brm with RNA polymerase II (RNAPolII) is one such mechanism [88]. In this case, Brm acts as a signal transducer of the MAP Kinase pathway and promotes alternative splicing of E-cadherin (*CDH*1), apoptotic regulator BCL2-like 11 (*BIM*), cyclin D1 (*CCND*1) and *CD44* [88]. Altered splicing of *CD44*, *FGFR2*, *RAC1* or *MST1R* has been shown to promote invasive potential in cancerous cells [89].

Exon-skipping on certain cancer-implicated genes has also been correlated with increased levels of H3K9me3 and recruitment of heterochromatin protein isoform HP1γ [90]. These include Glutaminase 1 (*GLS1*), BRCA1 DNA Repair Associated (*BRCA*1), DSN1 Component of MIS12 Kinetochore Complex (*DSN*1), and Protein Kinase N2 (*PKN*2) genes. Another study has shown that the enrichment of HP1α isoform and facultative heterochromatin marks (H3K9me2 and H3K27me3) at the fibronectin gene (*FN*1) causes variant exon inclusion [87]. During EMT in non-small cell lung cancer (NSCLC) TGF-β induces overexpression of an alternatively spliced Osteopontin isoform (OTNc). Splicing of OPTc is HDAC dependent and enhanced by the RUNX2 transcription factor [91]. Recent studies have concluded that the higher expression of OPTc also provides resistance to cisplatin treatment through activation of Ca^2+^/NFATc2/ROS signaling [92].

## 5. Impact of Chromatin Disorganization on Cancer Progression

### 5.1. Diversification of Tumors by Mislocalized or Disorganized Chromatin

An important aspect of cancer progression and relapse is the inherent heterogeneity of tumors. High-throughput mapping of the different regions of tumors has demonstrated the presence of at least 4–8 distinct subclonal populations within the tumor [93]. This intra-tumor heterogeneity is further enriched by the chromatin rearrangements induced during stochastic tumor proliferation or metastasis. As discussed in earlier sections, these rearrangements can alter the expression of genes by novel fusion transcription factors, hijacking of distal regulatory elements, or changes in epigenetic marks (Table 2).

In addition to the altered chromatin organization and function, differential intranuclear positioning of chromosomes (e.g., chr18 and chr19) has also been observed in cancer cells both in vitro and in tumor samples [138,139,140]. Such evidence raises two possible roles of the altered nuclear organization in cancer: altered gene expression by repositioning the genes to a different nuclear compartment, or due to the accelerated mutational/translocation rates in the new nuclear compartment. Nuclear lamina-associated chromatin is transcriptionally repressed, and repositioning to the nuclear lamina has been shown to attenuate gene expression in several model systems [141].

Another impact of the repositioned chromosomes is on the mutation rates. In agreement with the protective role of the constitutive heterochromatin proposed five decades ago, it has been observed that the peripheral chromatin has a higher somatic mutation frequency in cancer cells than the corresponding chromatin at the nuclear interior [142]. The higher incidence of mutations in the periphery-associated chromatin suggests a combination of factors such as higher exposure to external mutagens and predominant error-prone repair mechanisms. A similar correlation of high-mutational rates in heterochromatic domains of diverse types of cancers had been made earlier, where it was reported that ~40% of the somatic mutations were found in H3K9me3 enriched heterochromatin [143]. Further, it has been observed that certain chromosomal translocations in cancer are more frequent than others. In normal nuclei, these chromosomal loci are found to be positioned close to each other, thereby raising the probability of chromosomal fusions between them during DNA repair. Such events are expected to be more prevalent in the heterochromatic regions of the genome, where microhomology-mediated end-joining (MMEJ) is the preferred mode of DNA repair [144].

In certain cancers, an increase in the heterogeneity of tumors is also attributed to the presence of extrachromosomal DNA (ecDNA) particles. These ecDNA lack the higher-order compaction typically seen on chromosomes [143]. This accessible, open chromatin conformation permits ultra-long-range chromatin contacts driving high expression of oncogenes, often correlating with poor prognosis in multiple types of cancer [145].

### 5.2. Aneuploidy and Evasion of Therapeutic Interventions

Cancer cells are subject to multiple endogenous and exogenous factors that cause chromosomal numerical instability (CIN), contributing to tumor heterogeneity. The genetic heterogeneity of cancer cells is analogous to the heterogeneity observed in asexual unicellular organisms such as yeasts. It has been observed that tetraploid yeasts evolve faster than their diploid counterparts in response to extracellular stresses such as nutrient deprivation [146]. However, the effects of change in ploidy are context dependent: imbalanced gene expression in certain aneuploid cells can lead to fitness penalty and reduced growth potential, while under stressed conditions, the aneuploid colorectal cancer cells or even non-transformed human fibroblasts have a better proliferative potential [147]. A similar aneuploidy-induced growth advantage has been reported for the antifungal drug-resistant Candida albicans, thiol peroxidase-deficient budding yeast, and serum-starved colon epithelial cells [147]. Therefore, it is very likely that the polyploidy in cancer cells may confer similar adaptability to cells stressed by hypoxic conditions within the tumor microenvironment or by therapeutic pressures.

Further, the effect of a spontaneous deleterious mutation or rearrangement can be minimized by the presence of a gene in multiple copies. This effect, commonly known as Muller’s ratchet, was originally proposed to predict the evolutionary outcome of the accumulation of irreversible deleterious mutations. This phenomenon is also bolstered by observations that complex genomic rearrangements such as chromothripsis are frequent in aggressive tumors [124,148]. Such a rearrangement may lead to the disruption of genes, the creation of novel fusion oncogenic proteins, or the amplification of certain oncogenes on the mosaic chromosome. However, only a small fraction of cells in the tumor survive the rearrangements at such a massive scale. The surviving cells may contribute to the diversification of the tumor cell repertoire. The persistence of a smaller subset of spontaneously formed “drug-tolerant” cancer cells with a distinct chromatin structure has also been implicated in clonal evolution as a mechanism of resistance to chemotherapies [149].

### 5.3. Increased Invasive Potential of Tumors

Migration of cells through tissue microenvironments occurs in a variety of functional contexts and is essential for development, immune surveillance, and tissue morphogenesis. The migration is enabled by the positioning and shaping of the nucleus and involves two independent processes: transcription of the cytoskeleton-reorganizing genes and compaction of chromatin. The role of chromatin compaction in migration is evident from a study where drug-induced chromatin decondensation led to reduced cell migration in a transcription-independent manner [150]. Increased chromatin compaction in response to migration cues has also been observed in filamentous fungi, *Neurospora crassa* [151]. Further, during the cellularization of Drosophila embryos, the elongation of nuclei coincides with the condensation of chromatin to a distinct chromocenter [152]. Such evidence indicates that chromatin compaction regulates nuclear shape mechanics and stiffness during migration (Figure 4).

Dysregulated chromatin compaction in disease can be best exemplified through the heterochromatin protein (HP1) family of non-histone chromatin proteins. Human cells contain three conserved homologs of HP1 proteins (HP1α, HP1β, and HP1γ) that are recruited to the repressive H3K9me^2,3^ marks by N-terminal chromodomain, and to other binding partners by C-terminal chromoshadow domain [128]. The reduced levels of HP1α isoform or loss of its dimerization property have been shown to promote the metastatic ability of breast cancer cells by disrupting the tethering of peripheral heterochromatin to the nuclear lamina and making the nuclei more malleable [129,130,131]. In the absence of HP1α, the interphase or mitotic chromosomes lose rigidity, display higher chromosomal segregation errors, and cause abnormal nuclear morphology [153]. This HP1α-mediated chromatin stiffness is induced by the bridging of chromatin fibers, independent of the stiffness induced by an increase in global histone methylation levels, demonstrating at least two distinct pathways that regulate chromatin mechanics. Therefore, it is of no surprise to find that the HP1α cells have no significant change in gene expression, local chromatin compaction, or histone methylation marks [153]. Summarily, it can be envisaged that chromosomal rigidity and mechanics are key aspects in maintaining nuclear stiffness and in controlling the metastatic potential of tumor cells (Figure 4). Interestingly, even the increased expression of HP1 isoforms (HP1α and HP1β) has been shown to have a detrimental effect on sister-chromatid cohesion and telomere length, ultimately leading to end-to-end chromosomal associations and damage [154,155].

It has also been observed that increased chromatin compaction induced by osteopontin-mediated signaling enables bone marrow-derived mesenchymal stem cell (BMSC) nuclei to overcome deformations during migration [156]. The cellular migration-induced chromatin condensation appears to be a recurrent theme, coinciding with the reduced nuclear volume and improved resistance to nuclear deformation during tumor cell migration through interstitial tissues [157,158]. It has been proposed that physical bridging between the cytoskeleton and condensed chromatin enables coordinated structural changes in cellular shape for directed migration [158]. This argument is favored further by observations that the volume of isolated nuclei reversibly increases up to two-fold when chromatin decondensation is induced by the chelation of divalent cations [159]. Nuclear structure and chromatin compaction are also altered by the over-expression of transcriptional co-activator (p300) and RET/PTC oncogenes [160,161], suggesting that other oncogenes may also alter metastatic potential through nuclear deformation.

## 6. Targeted Cancer Therapeutics Aimed at Chromatin Modifiers

DNA-damaging chemotherapeutic agents can have an unintended effect on normal cells and promote further heterogeneity in the cancerous cells through chromosomal rearrangements. Increased phenotypic plasticity with destabilized chromatin has been proposed to provide a selective advantage to invasive tumor cells [159] (Figure 4). Therefore, targeting chromatin per se rather than inducing DNA damage may have higher success potential as a therapeutic approach for the treatment of cancer and other complex diseases. The reversibility of the chromatin modifications and their involvement in distinct cancer subtypes have led to trials of chemical inhibitors aimed at the epigenetic modifiers (Table 3). A few of the prominent therapeutic approaches targeting chromatin in cancers are listed here.

### 6.1. Direct Inhibition of Epigenetic Modifiers

The largest subgroup of epigenetic inhibitors approved by the US Food and Drug Administration (FDA) for clinical applications are inhibitors of DNMT (e.g., Vidaza, Dacogen) and HDACs (e.g., Zolinza, Istodax, Beleodaq, Epidaza) [102] (Table 3). These two approved DNMT inhibitors are cytosine analogs. At low concentrations they cause loss of methylation marks during DNA replication, thereby promoting tumor suppressor gene activity. Antitumor activity of these inhibitors has also been attributed to induced G2 arrest due to the accumulated DNA double-strand breaks and the activation of interferon response to the expressed retroviral elements [162]. However, being a substrate for cytidine deaminase, these nucleoside analogs have a low in vivo stability.

The HDAC inhibitors, on the other hand, restore the acetylation status of several histone and non-histone proteins and promote the expression of tumor-suppressor genes. For instance, the HDAC inhibitor Zolinza (Suberoylanilide hydroxamic acid or SAHA), approved for the treatment of cutaneous T cell lymphoma (CTCL) patients, is an effective inhibitor of HDAC I, II, and IV [102]. It has also been shown to induce cell-cycle arrest and apoptosis and to resensitize lymphoma cells to chemotherapy [162]. However, these compounds have significant side effects and are ineffective against solid tumors. Unlike the class I or II HDACs, inhibition of the class III HDAC (SIRT1) has been found to reactivate the expression of several tumor-suppressor genes, even when the promoter of these genes is hypermethylated [148]. This unique response to SIRT1 inhibition has a potential clinical application in restoring the expression of abnormally silenced tumor suppressor genes.

The selective inhibition of other epigenetic writers and readers has also proved to be promising in clinical oncology. The targets of these drugs include lysine-specific demethylase 1 (LSD1), enhancer of zeste homolog 2 (EZH2), and bromodomain and extra-terminal motif (BET) proteins. The BET family proteins are of particular interest due to their ability to bind to acetylated lysines to stimulate transcriptional activity and as determinants of epigenetic memory [165]. The most studied member of this family is BRD4. The formation of a fusion oncoprotein BRD4-NUT due to a chromosomal translocation t(15;19) results in an aggressive form of squamous carcinoma, commonly referred to as NUT midline carcinoma [166]. Mechanistically, this BRD4-NUT fusion protein recruits the histone acetyltransferase p300 and establishes hyperacetylated megabase-sized chromatin domains that function as super-enhancers for *MYC*, *SOX*2, and *TP*63 expression [166]. This has led to a targeted search of acetyl-lysine mimetics as BET inhibitors (iBETs) for anti-cancer properties (Table 3). Notable examples of such inhibitors are JQ1 and iBET762, which reversibly bind to bromodomains of BET proteins and evict them from chromatin, resulting in the differentiation and apoptosis of cancerous cells. Activating mutations on the EZH2 catalytic component of PRC2 cause hypermethylation of H3K27 residues and have been identified in B-cell lymphoma and non-Hodgkin’s lymphoma patients. Successful use of EZH2 inhibitors in these patients has shown antiproliferative effects on lymphomas. The first clinically approved EZH2 inhibitor is Tazemetostat for the treatment of locally advanced or metastatic epithelioid sarcomas [167].

### 6.2. Cancer-Associated Metabolic Enzymes and Metabolites

Certain metabolites such as acetyl coenzyme A (acetyl-CoA), S-adenosylmethionine (SAM), α-ketoglutarate (αKG), and lactate act as cofactors or substrates of the chromatin-modifying enzymes. This metabolite dependency of the epigenetic modifiers makes the metabolic enzymes potential targets in cancers to achieve indirect modulation of chromatin. A classic example is isocitrate dehydrogenase (IDH) which catalyzes the synthesis of αKG. Missense mutations in *IDH*1/2 have been frequently observed in gliomas and acute myelogenous leukemia (AML), increasing the metastatic potential [168,169]. The neomorphic IDH mutants cause the conversion of αKG to an oncometabolite R-2-hydroxyglutarate (R-2HG) instead of isocitrate. At physiological concentrations, R-2HG is inhibitory to the activity of the ten-eleven translocation (TET) methyl-cytosine hydroxylases and JmjC domain-containing histone demethylases [170]. Targeted inhibition of the mutant IDH1, but not the wild type IDH1, through AGI-5198 treatment has been found to arrest the growth of gliomas [171]. Enasidenib is the first-in-class mutant IDH inhibitor approved for the treatment of AML. Another drug, Ivosidenib, has also been approved for AML and the drug is in advanced trials for cholangiocarcinoma (ClinicalTrials.gov Identifier: NCT02989857).

S-adenosylmethionine (SAM) is the methyl donor used by DNMTs, lysine, and arginine methyltransferases. The depletion of SAM has been observed in several cancers due to overexpression of the enzyme Nicotinamide N-methyltransferase (NNMT) [172]. The enzyme catalyzes the transfer of methyl moiety from SAM to nicotinamide, producing 1-methyl-nicotinamide (MNA). The metabolite MNA acts as a stable sink for methyl-residues, leading to the hypomethylation of histones and signaling proteins such as tumor suppressor protein phosphatase 2A (PP2A) [172]. Owing to this role, NNMT is an established cancer-associated metabolic enzyme and its overexpression has been implicated in tumor progression, metastasis, and poor clinical prognosis [173]. The use of mimetics targeting either SAM or Nicotinamide-binding pocket of the NNMT has recently gained interest as a viable therapeutic approach. NNMT inhibitors derived from methylquinolinium (MQ) such as 5-amino-1MQ, 7-amino-1MQ, and 2,3-diamino-1MQ have been reported to reduce tumor cell proliferation and improve H3K27 trimethylation levels. However, their stability and efficacy in pre-clinical in vivo cancer models are still being investigated [174].

### 6.3. Other Chromatin Modulators in Cancer Therapeutics

As mentioned earlier, conventional chemotherapy by DNA damaging agents may contribute to the heterogeneity of tumor cells, irreversibly damaging normal cells. These limitations can be overcome by using chromatin disruptors that intercalate between DNA bases without causing any chemical modification. One such class of drugs in clinical trials is Curaxin (e.g., CBL0137, ClinicalTrials.gov Identifier: NCT03727789), which destabilizes nucleosomes by intercalating between major and minor grooves on DNA, altering the negative charge, contour length, and bending rigidity of the DNA. This destabilization of nucleosomes or “chromatin-damage” causes the trapping of histone chaperone FACt (FAcilitates Chromatin Transcription) on the DNA [159], thereby indirectly exhausting the cellular pool of FACT. This sequestration of the chaperone FACT stabilizes and activates tumor-suppressor p53. Curaxin also activates type I interferon response against the transcription of centromeric heterochromatin and attenuates expression of MYC, NF-kB, and HSF1 responsive genes by disrupting the long-range chromatin interactions [159]. Cumulatively, these phenotypic changes are detrimental to the viability of tumor cells.

Similarly, the use of chemicals that bind weakly to DNA or DNA-binding proteins (e.g., topoisomerases) has also been found to have anticancer properties. Efficient DNA damage repair in response to radiation therapy has been reported as a key survival strategy for cancerous cells [175]. These observations have prompted the development of specific inhibitors of the DNA repair pathway. The use of small molecule activators that can reverse the effect of over-expressed cancer-associated protein is also being investigated. The foremost examples of such activators are the MDM2 agonists (RG7388 and HDM201) which block the interaction between MDM2-p53 and rescue tumor-suppressor activity of p53 [176].

As oncohistones disrupt HMTase activity and lead to altered expression of certain regulators of mesenchymal differentiation, homeostasis can be restored by either gene-editing to eliminate mutant histone gene or by a therapeutic intervention of oncogenic driver gene (e.g., WEE1 kinase inhibitor, AZD1775 for SETD2 deficient cells). The histone methyltransferases recruit DNMTs to gene promoters creating a lock for permanent silencing. This permanent silencing can be reversed by the combined recruitment of histone deacetylases (HDACs) and DNA demethylases [177,178]. On the contrary, inhibition of DNA methylation alone can restore the expression of genes to near-normal levels. Based on these results, a combinatorial regimen of DNA methylation and HDAC inhibitors has been proposed for myeloid neoplasms [179]. Interestingly, targeted inhibition of the methylcytosine-binding proteins (MBPs) that act downstream of the DNA methylation process can have a similar effect on gene activation without the removal of DNA methylation marks. SIRT1, a class III HDAC inhibitor, also achieves the same result of gene activation by bypassing the hypermethylated promoters [180]. Therefore, it appears that the primary role of epigenetic marks is to alter the chromatin compaction state, and any pharmacological intervention or loss-of-function mutations with the potential to increasing the accessibility to genes can promote their expression.

### 6.4. Targeted Degradation of Proteins

A radically different therapeutic modality is targeted protein degradation using proteolysis-targeting chimeras (PROTACs) or related small-molecule drugs [181]. In this approach, a highly selective hetero-bifunctional ligand recruits the target protein to an E3 ubiquitin ligase. This causes the target protein to be polyubiquitinylated and degraded by the ubiquitin-proteasomal system. Compared to the broad-spectrum pharmacological inhibitors, the PROTAC system provides target-selectivity of the bifunctional ligands and has reduced off-target effects. Unlike the small-molecule inhibitors that compete for binding to the active site to achieve inhibition, the PROTACs can induce degradation by targeting any region of the protein. This enables activity against “undruggable” proteins with no known inhibitor.

However, the approach faces several unique challenges before it can be tested clinically. Currently, a very small fraction of ~600 known E3 ligases have been studied for targeted degradation [181]. Furthermore, it was found that among the known E3 ligases, only 24 have a ubiquitous expression [181]. Therefore, other alternative E3 ligases need to be carefully evaluated for their tissue-specific expression and pharmacokinetics. Another option is the simultaneous and controlled expression of targeting E3 ligases through transgenesis in patient-derived induced pluripotent cells (iPSCs) for therapies [182]. Acquired mutations in the target protein to the ubiquitin-proteasomal pathway components can also limit the applicability of the PROTAC modality in patients. However, despite these challenges, there are several PROTACs in developmental or clinical trials for cancer therapies. The first molecule of this class ARV-110 targets androgen receptors and is in clinical trials for the treatment of metastatic castration-resistant prostate cancer (ClinicalTrials.gov Identifier: NCT05177042). A few PROTAC molecules selectively targeting chromatin modifiers in cancer are FHD-609 (against BRD9, a subunit of non-canonical BAF complex) for the treatment of synovial sarcoma, and FHD-286 (against BRG1 and BRM chromatin remodelers) to treat acute myeloid leukemia and metastatic uveal melanoma (ClinicalTrials.gov Identifier: NCT04879017).

## 7. Conclusions

Chromatin assembly is essential not only for packaging of the genome in a confined nuclear volume, but also for controlled gene expression, cell differentiation, and response to extracellular cues. The hallmark features of heterogeneous gene expression and uncontrolled proliferation of cells observed at the onset of cancer are probabilistic outcomes of changes in chromatin organization. Therefore, the role of chromatin organization in cancer needs to be viewed as a primary determinant of pathology and worsening of prognosis rather than a secondary effect of the oncogene expression. The establishment and progression of cancer involve changes in chromatin at multiple levels either independently or in conjunction with other neoplastic events. To lend genome-wide perspective to the role of chromatin in oncogenesis, we are paraphrasing these events in two distinct levels.

Chromosomal alterations in cancer: Organization of chromosomes in euchromatic (A-compartment) and heterochromatic (B-compartment) chromatin is dynamic, and changes during cell differentiation. Frequently, cancerous cells also undergo transitions in this compartmentalization to repress the expression of tumor-suppressor genes or promote the expression of proto-oncogenes. Recent genome-wide studies on cervical cancer, triple-negative breast cancer, or neoplastic B-cells have demonstrated that at least a quarter of this compartmentalization is altered during neoplastic transformations [71,72,73]. As discussed in the earlier sections that the increased chromatin dynamics by these switching events also disrupt interactions between distal regulatory elements or TAD boundaries. The malignant impact of such switching of chromatin compartments is evident during the formation of cancer-specific TADs in prostate or breast cancer cells [70,73].

Nuclear organization and cancer: Cancerous cells are typically characterized by abnormal nuclear size and morphology. Expectedly, loss of inner nuclear membrane proteins such as lamins, disorganized heterochromatin, and chromosomal aneuploidies have been attributed to these morphological changes. However, the dependence of nuclear volume on the amount of compacted chromatin has been underappreciated. Increased expression of several oncogenes (e.g., p300 and RET/PTC) has also been observed to alter nuclear volume [160,161]. A critical outcome of these changes is evident in the repositioning of chromosomes within the nucleus of cancerous cells. Repositioning of chromosomes 4, 12, 15, 16, and 21 towards the nuclear periphery in breast cancer has been correlated to reduced expression of the genes resident on these chromosomes [140]. Similar reorganization of chromosomes 4, 9, 14, and 18 to nuclear interior in human myeloma nuclei has been documented too [139]. Reduced chromatin compaction leads to a loss of nuclear rigidity and an increase in nuclear blebbing and chromatin mobility. These aspects may serve as secondary oncogenic events by facilitating more chromosomal translocations and destabilization.

The dysregulated gene expression enables cancer through a two-pronged mechanism: increased diversity of cancerous cells promoting oncogenesis and inducing chromosomal instability, leading to more imbalance in gene expression. Currently, the biggest challenge in oncology is overcoming the resistance to targeted therapies and inducing sensitivity to immunotherapies with minimal damage to normal cells. In general, pharmacological interventions for specific vulnerabilities in cancer are effective against gain-of-function or cancer-specific chimeric proteins. In addition to the targeted epigenetic inhibitions, the combinatorial regimen has proven a more effective means to treat patients than the chemotherapeutic drugs or radiation treatments alone. Chromatin remodelers and associated factors are a promising avenue for exploration and therapeutic interventions, many of which are in late-stage clinical trials. It is in this area where a better understanding of nuclear architecture and chromatin organization will provide new paradigms to cancer research, diagnostics, and therapeutics.

## Figures and Tables

**Figure 1 cancers-15-00466-f001:**
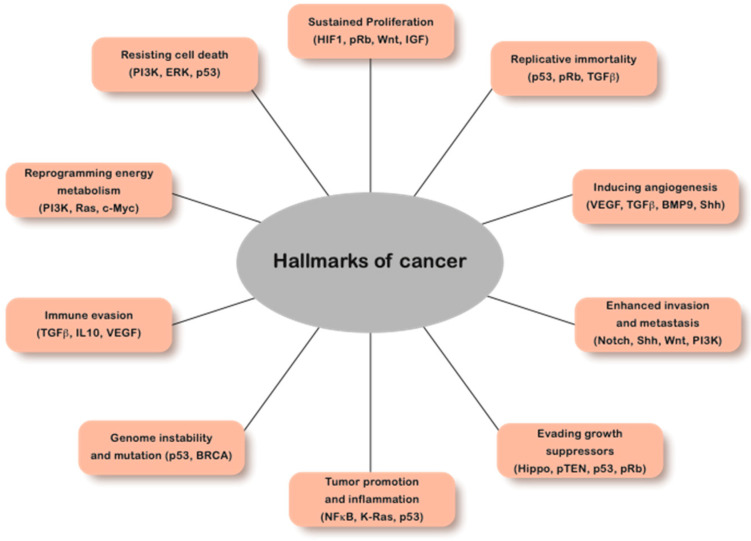
Hallmarks of cancer. A schematic representation of the pathways conferring growth advantage to tumor cells in different types of cancer. The prominent proteins of these pathways identified in patient samples are shown in brackets. Importantly, the tumor cells gain growth advantages through various mechanistic strategies and deregulated pathways. Source: [2,3,4,5,6,7,8,9,10].

**Figure 2 cancers-15-00466-f002:**
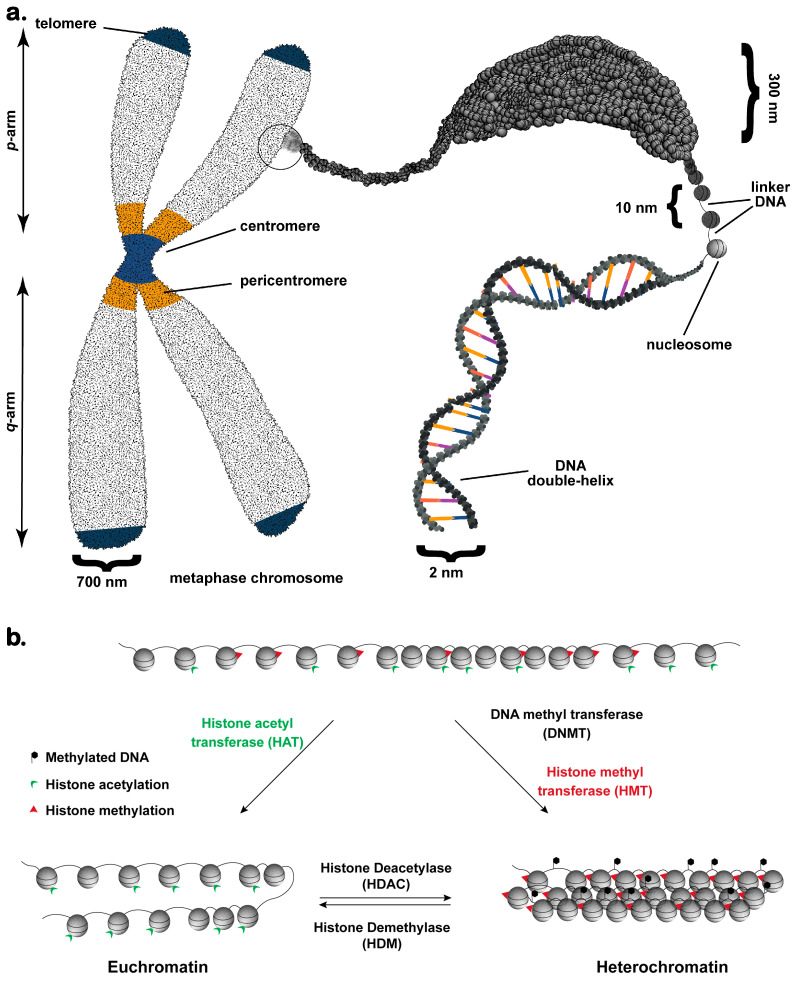
A schematic of chromatin organization, compaction, and remodeling by epigenetic modifiers in a metazoan chromosome. (**a**) The metaphase chromosome contains distinct regions essential for chromosomal integrity and segregation during mitosis. The chromatid is a higher-level folded structure containing 300 nm fibers (**top right**). The 300 nm fibers are the secondary level of compaction, which is composed of 30-nanometer chromatin fiber. The DNA–histone complex called nucleosomes is the fundamental unit of this chromatin fiber that folds sequentially to produce the 30 nm chromatin. (**b**) A schematic representation of chromatin organization as euchromatin (**bottom left**) and heterochromatin (**bottom right**) from the bivalent chromatin (**top**). The euchromatin is accessible to the chromatin remodelers, transcription factors, and polymerase complex. The DNA is largely unmethylated, and histones carry specific acetylation marks (green). On the contrary, the heterochromatin is decorated by methylated DNA (black hexagons), and methylated histones (red triangles). A discrete set of molecular erasers can remove these marks to alter the chromatin from one type to another. For clarity, an overview of chromatin organization is shown, and the diverse posttranslational modifications are not shown.

**Figure 3 cancers-15-00466-f003:**
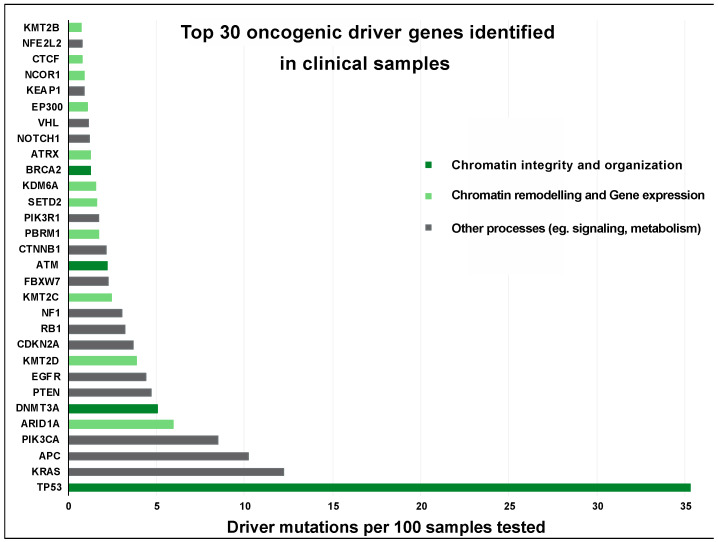
The oncogenic potential of the genes regulating chromatin organization and function reported in the clinical samples. A comparative histogram depicting the driver mutations identified in the top 30 genes involved in cancers. Source: 10 Pancancer studies available at the cBioportal (*n* = 76,639 samples) [58,59].

**Figure 4 cancers-15-00466-f004:**
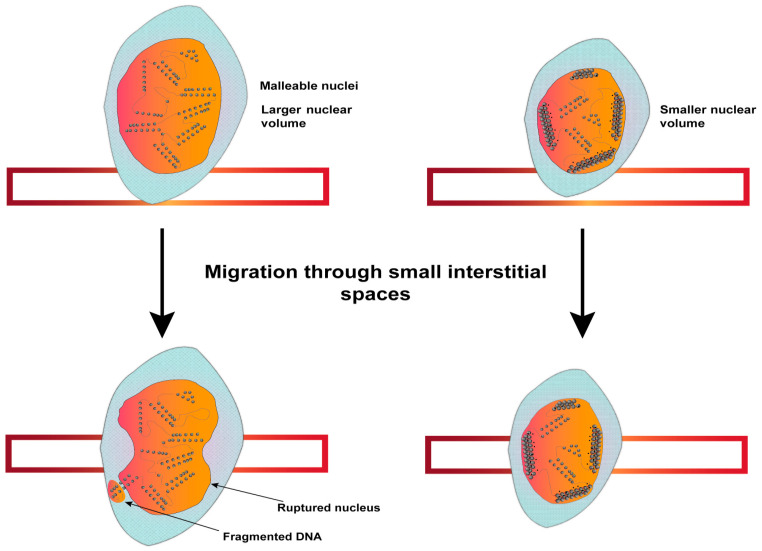
Role of chromatin compaction in maintaining nuclear integrity during metastasis and cellular migration. A schematic of cells migrating through small interstitial spaces in tissue microenvironments with decondensed (**left**) or compacted (**right**) chromatin. The nuclei with decondensed chromatin are larger in volume and prone to ruptures in the nuclear membrane during migration. Higher nuclear rigidity by the compacted chromatin redistributes the shearing stress and resists nuclear membrane ruptures.

**Table 1 cancers-15-00466-t001:** Inherited pathologies due to nuclear or chromatin disorganization and their association with cancer development.

Inherited Pathology	Involved Tissue	Mutant Protein/RNA	Mechanism	References
Abnormal nuclear morphology	Renal cell carcinoma (RCC), gastrointestinal cancers, breast cancer, and cervical cancer	Depletion of AT-rich interactive domain 1A (*ARID1A*)	Increase in nuclear volume, cell proliferation, migration, and chemoresistance	[36]
Abnormal nuclear morphology	Melanoma, and bladder cancer	Loss of macroH2A1 and macroH2A2 histone variants	Defects in nuclear organization, including disruption of nucleoli and a global loss of dense heterochromatin	[37]
Abnormal nuclear morphology	Human cervical cancer	Reduction of NOP53 ribosome biogenesis factor (*NOP53*)	Increased chromosomal instability, multinucleated cells, nuclear budding	[38]
ICF syndrome	Immunodeficiency due to reduced or absent serum immunoglobulins, facial abnormalities, and developmental delay	DNA Methyltransferase 3B (*DNMT*3B), Zinc-finger & BTB domain containing 24 (*ZBTB*24), Cell division cycle associated 7 (*CDCA*7) or Helicase, lymphoid-specific (*HELLS*)	Hypomethylation of satellite repeats at the pericentromeric heterochromatin activating interferon-mediated innate immune response	[39]
Laminopathy	Small-cell lung cancer, prostate cancer, pancreatic cancer, and melanoma	Altered *LMNB*1/2 expression	Epigenetic derepression of the *RET* proto-oncogene by loss of PRC2 recruitment	[31,40,41]
Laminopathy	Breast Cancer, colorectal cancer, melanoma, gastric cancer, leukemia, and lymphoma	Mutations or reduced expression of *LMNA*	Destabilization of retinoblastoma (pRb) or hyperactivation of MAPK, PI3K/AKT pathways	[35,42,43]
Lynch syndrome hereditary non-polyposis colorectal cancer (HNPCC)	Colorectal, ovarian, and endometrial cancers	Epimutation (Deletion in *TACSTD*1) or mutation in associated genes (*MLH1*, *MSH2*, *MSH6, PMS2* and *EPCAM)*	Mosaic and allele-specific hypermethylation of the downstream *MSH*2 promoter	[44]
Nuclear envelopathies	Ovarian cancer, prostate cancer, lung cancer, breast cancer, colorectal cancer	Emerin, Nesprin-1, and Nesprin-2	Aneuploidy & chromosomal numerical instability. Altered chromatin conformation reduces *GATA6* expression	[45,46,47]
Pelger-Huët anomaly and Greenberg Dysplasia	Multilobed, hypo-segmented nuclei form in white blood cells. Increased LBR expression is seen in aggressive breast cancers	Lamin B Receptor (*LBR*)	Mislocalization of inactive X (Xi) to the nuclear interior causes its genes to express	[40,48,49]
Thalassemia	Hepatocellular carcinoma	Epimutation (Deletion in L*UC7*L gene)	*HBA*2 gene silencing induced by promoter hypermethylation	[50,51]
Werner’s Syndrome, Aicardi-Goutières syndrome	Colorectal adenocarcinoma, metastatic prostate cancer, leukemia, cervical cancer, and ovarian cancer	Reduced expression of RNaseH2A	Genomic instability, increased metastasis, cellular senescence, ageing symptoms	[52]

**Table 2 cancers-15-00466-t002:** Role of chromatin modifiers in cancer establishment and progression.

Chromatin Modification	Gene/Region Involved	Known Cancer Association	References
DNA Methylation			
Promoter Hypermethylation	*RASSF1*	Hepatocellular carcinoma, oral squamous cell carcinoma, lung, breast, colorectal, bladder, cervical, and prostate cancers	[81,83,94,95,96]
	*CDH1*	Prostate cancer, hepatocellular carcinoma, Non-small cell lung carcinoma (NSCLC), esophageal, gastric, breast and bladder cancers	[81,83,97]
	*DAPK1*	Breast, cervical, and bladder cancers	[96,97,98]
	*CDKN2A*	Melanoma, glioblastoma, bladder cancer	[81]
Hypomethylation	*HOX11*	Leukemia	[99]
	*pS2*	Breast cancer	[99]
	c-n-RAS	Most adult cancers	[99]
	*C-MYC*	Colorectal cancer	[99]
	LINE-1 repeats	Prostate cancer	[83]
Genebody Methylation	p53-exon5	Non-small cell lung carcinoma (NSCLC)	[100]
	*HIF-1α*	Breast cancer	[101]
**Histone modifications**			
Methylation	*EZH2*	Most adult cancers	[99,102]
	*KMT2D*	Breast cancer	[102]
	*SETD2*	Renal cell carcinoma, Lung cancer	[99,102]
Acetylation	*E2F1*	Colon cancer	[103]
	*Mcl-1*	Chronic myeloid leukemia (CML)	[104]
	*Ku70*	Neuroblastoma, hepatocellular carcinoma	[105]
	*EP300*	Breast, colorectal, pancreatic cancer	[106]
	*HDAC2*	All major cancers	[102,107]
Phosphorylation	*PKC*	Chronic Lymphocytic Leukemia (CLL), colorectal carcinoma, melanoma, invasive ductal breast cancer, NSCLC	[108,109]
	*ATM/ATR*	Epithelial, breast, and pancreatic cancers, leukemias, lymphomas	[110,111]
	H3tyr41	Leukemia	[112]
	Aurora B	Breast and colorectal cancers	[113]
Chromatin Remodeling/pre-mRNA Splicing	*ARID1A*	Colon cancer, ovarian clear cell cancers, uterine endometrial cancers, renal cell carcinoma	[36,114,115]
	*BRCA1*	Breast and ovarian cancer	[116]
	*BRM*	Prostate cancer, basal cell carcinoma, Lung cancer	[117,118]
	*CHD4/5*	NSCLC, Colorectal, gastric, ovarian, and Prostate cancers	[119,120]
	*ASXL*	Myelodysplastic syndromes, acute myeloid leukemia (AML)	[114]
**Structural changes**			
Loss of heterochromatin	Barr body	Breast cancer, Ovarian cancer	[13,53,54]
	Pericentromeric and telomeric heterochromatin	Most adult cancers, Lung cancer	[121,122]
Rearrangements	Genomewide local clustered rearrangements/Chromothripsis	Sonic-Hedgehog medulloblastoma, AML, aggressive tumors	[123,124]
	Satellite repeats	Colorectal cancers	[125]
	*TET1*	Osteosarcoma, AML	[99]
	*BRD4*	Midline carcinoma, breast and colon cancer, AML	[126,127]
Chromatin conformation and stiffness	HP1α	Breast cancer	[128,129,130,131]
	*GATA3*	Acute lymphoblastic leukemia	[132]
	*IDH1/2*	Glioma, Chondrosarcoma, Cholangiocarcinoma, Myelodysplastic syndrome (MDS), AML	[133]
	*STAG2, RAD21, SMC1A* and *SMC3* (Cohesin complex)	Myeloid leukemia, Breast cancer, Lung adenocarcinoma	[134]
**Long-Range interactions**			
Enhancer hijacking	*MYB*	Adenoid cystic carcinoma	[65]
	*BCL11B*	Lineage-ambiguous leukemia	[66]
	*KLF5*	Head and neck squamous cell carcinoma, esophagial carcinoma	[14,67,135]
Super-Enhancer deletion	*RCAN1.4*	Breast cancer	[68]
Enhancer Focal amplification	*MYC*	Lung adenocarcinoma, endometrial carcinoma	[67]
	*PARD6B*	Liver hepatocellular carcinoma	[67]
	*USP12*	Colorectal cancer	[67]
TAD disruption	*AR, FOXA1*	Prostrate cancer	[70]
	*PDGFRA*	Glioma	[136]
	*TAL1*	T cell acute lymphoblastic leukemia	[14,137]
	*LMO2*	T cell acute lymphoblastic leukemia	[14,137]

**Table 3 cancers-15-00466-t003:** List of epigenetic inhibitors and their approval status for clinical applications.

Inhibitor Category	Prominent Examples	Generic Name of FDA Approved Drug	Brand Name and Manufacturer	Therapeutic Use
Acetylated Histone binding protein inhibitor (PAHi)	CPI203, RVX-208, I-BET-726	-	-	-
Bromodomain (BRD) and extra-terminal domain (BET) protein inhibitor (BETi)	OTX15, I-BET762, I-BET151, JQ1, Pelabresib (CPI-0610), Molibresib (GSK525762), INCB054329, INCB057643, ODM-207, Ten-010 (RO-6870810), BAY 1238097, SF-1126, Trotabresib (CC-90010), AZD-5153, PLX-51107	**Nivolumab** (BMS-986158)	**OPDIVO**^**®**^ by Bristol-Myers Squibb Pharma, NY, USA	Advanced NSCLC, melanoma, renal cell carcinoma, squamous cell carcinoma, hepatocellular carcinoma, urothelial carcinoma, colorectal cancer, classical Hodgkin’s lymphoma, malignant pleural mesothelioma
DNA Methyl Transferase inhibitor (DNMTi)	Epigallocatechin-3-gallate, Zebularine, Equol, Genistein, Guadecitabine (SGI-110), Procaine, Nanaomycin A, Disulfiram, Lomeguatrib, RG108, SGI-1027, MG98, CP-4200, Hinokitiol, DC_517, DC-05, Isothiocyanate, Fazarabine (Arabinosyl-5-azacytidine), DHAC (5,6-dihydro-5-azacytidine)	**Decitabine (5-aza-2′deoxycytidine)** **5-Azacytidine** **Procainamide**	**Dacogen**^**®**^ by MGI Pharma, Inc., NJ, USA **Vidaza**^**®**^, **Onureg**^**®**^. Both by Bristol-Myers Squibb Pharma, NY, USA **Pronestyl**^**®**^ by Nicholas Piramal India Ltd., Mumbai, India and Bristol-Myers Squibb Pharma, NY, USA	Myelodysplastic syndrome (MDS) Myelomonocytic leukemia (CMML) Cardiac arrythmia
Histone Acetyl Transferase inhibitor (HATi)	Gallic acid, Garcinol, Anacardic acid, Procyanidin, MB-3, CTK7A, Plumbagin, Embelin, Curcumin, A-485, C646, DS17701585, Remodelin hydrobromide, Butyrolactone 3, CPTH2	-	-	-
Histone Deacetylase inhibitor (HDACi)	Givinostat, AR-42, Entinostat, Apicidin, Pracinostat, Abexinostat, Resminostat, CUDC-101, Toxoflavin, Inauhzin, Cambinol, Salermide, Trichostatin A, CG-1521, OSU-HDAC-42, HC-toxin, Plitidepsin, Tasquinimod, Sodium butyrate, Mocetinostat, Tefinostat, CHR-3996, QUISINOSTAT, Sodium phenylbutyrate, Pivanex, Butyroyloxymethyl-diethyl phosphate, Resveratrol, Dacinostat, Droxinostat, Psammaplin A, ITF-A, ITF-B, OSU-HDAC-44, Ricolinostat, Tubastatin A, RGFP966, TMP195, Fimepinostat, LMK-235, ACY-738, PCI-34051, Nexturastat A, CAY10603, ACY-775, WT-161, MC1568, RGFP109, Citarinostat, Scriptaid, Tucidinostat, Santacruzamate A, EDO-S101, Oxamflatin, HPOB, BML-210, Pomiferin, Domatinostat, BG45, Bufexamac, Sinapinic acid, FT895, CHDI-390576	**Vorinostat**, **Panobinostat** (LBH589), **Belinostat** (PXD101), **Romidepsin (FK228**, **Depsipeptide**), **Valproic acid**, **Valproic acid and divalproex sodium** **Carbamazepine**	**Zolinza**^**®**^ by Merck & Co., Inc., NJ, USA **Farydak**^**®**^ by Novartis, Basel, Switzerland **Beleodaq**^**®**^ by Acrotech Biopharma Inc., NJ, USA **Istodax**^**®**^ by Bristol-Myers Squibb Pharma, NY, USA **Stavzor**^**®**^ by Noven Pharmaceuticals, FL, USA **Depakene, Depakote** by Abbott Laboratories, IL, USA **Tegretol**^**®**^ by Novartis, Basel, Switzerland	Cutaneous T-cell lymphoma (CTCL) Multiple myeloma Multiple myeloma (discontinued) Elapsed or refractory peripheral T-cell lymphoma (PTCL) Cutaneous T-cell lymphoma (CTCL) Anticonvulsant Anticonvulsant (advanced-stage trials for breast cancer) Anticonvulsant
Histone Demethylase inhibitor (HDMi)	Pargyline, Clorgyline, Bizine, GSK2879552, KDM5-C70, JIB-04, ORY-1001, SID 85736331, Namoline, CBB1007, Methylstat, GSKJ4, GSKJ1, QC6352, SP2509, KDOAM-25, T-448, Daminozide, CPI-455, NCGC00244536, NCGC00247743, GSK-J2, Corin, GSK690, PBIT, S 2101, T-3775440 hydrochloride, INCB059872, CC-90011	**Tranylcypromine**, **Phenelzine**	**Parnate**^**®**^ by GlaxoSmithKline, Brentford, UK **Nardil**^**®**^ by Pfizer, NY, USA	Antidepressant (being investigated for anticancer properties) Antidepressant (Phase 2 trials for prostate cancer)
Histone Kinase inhibitor (HKi)	Ruxolitinib, KU-55933, VE-821	-	-	-
Histone Methyl Transferase inhibitor (HMTi)	UNC0321, UNC0224, EPZ-6438, DZNep, GSK343, Chaetocin, BIX-01338, BIX-01294, UNC0638, EPZ005687, GSK126, EPZ-5676, EPZ004777, SGC0946, E72, A-366, UNC1999, CPI360, UNC0965, BIX-01337, EI1, GSK503, BCI-121, LLY-507, EPZ015666, UNC0642, AZ505 ditrifluoroacetate, GSK3326595, MS023, JNJ-64619178, CM-579, EED226, MI-503, EPZ015866, MI-463, MI-538, MS049, CPI-169, BRD9539, LLY-283, EML741, OTS186935, SGC3027, Pinometostat	**Tazemetostat** (E7438/EPZ6438)	**Tazverik**^**®**^ by Epizyme, MA, USA	Advanced epithelioid sarcoma, follicular lymphoma
Methylated Histone binding protein inhibitor (PMHi)	UNC669, UNC1215	-	-	-
Poly (ADP-Ribose) Polymerase inhibitor (PARPi)	AMF-26, Talazoparib, Ilimaquinone, Veliparib, Niraparib, Rucaparib	**Olaparib** (AZD-2281)	**Lynparza**^**®**^ by AstraZeneca, Cambridge, UK and Merck & Co., Inc., NJ, USA	BRCA-mutated advanced ovarian cancer
Protein Arginine Demethylase inhibitor (PADi)	YW3-56, YW4-03, YW4-15, D-o-F-amidine, Cl-amidine, GSK484	-	-	-
Protein arginine methyltransferases (PRMTs) inhibitor (PRMTi)	AMI-1, Sinefungin	-	-	-
Ubiquitin Signaling Inhibitor (USi)	PTC209, GW7647, PRT4165, ML323	-	-	-

List of epigenetic inhibitors, and details of the FDA-approved drugs in respective categories for clinical applications. Inhibitors of different epigenetic modifiers are listed. The inhibitors approved for clinical applications by the US Food and Drug Administration are listed with their generic and brand names (in bold). Manufacturer information on the FDA-approved drugs is included with their brand names. Source [163,164], and https://clinicaltrials.gov (accessed on 20 November 2023).

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
