# Peer review of "Chromatin and Cancer: Implications of Disrupted Chromatin Organization in Tumorigenesis and Its Diversification"

_cancers, 2023, doi:10.3390/cancers15020466_

Round 1

Reviewer 1 Report

In this review, the authors have done a good job at compiling the existing literature associated with cancer. However, there is lot of improvement needed to make it useful to the concerned readers. Here are my suggestions:

1) The whole review needs more schematic representation of the pathways and concepts. In the current, its more like putting together the literature/references.

2) Table 1 needs better compilation. Would be good to have multiple columns like chemical name, manufacturer, approval status, reference, each in separate column.

3) Cut down the sections which are loosely related to cancer such as "Chromatin abnormalities and inherited pathologies". 

4) Even the introduction needs to be linked well with cancer. 

5) Would be good to have one more table showing the chromatin associated genes and its relevance in different types of cancer with few more annotations.

Reviewer 2 Report

In the section of conclusion: add a diagram to outline each specific "layer" of chromatin that has be discussed.  This will help lead to some specific prospectives as the end of this review. 

1) DNA mutation,

2) DNA methylation 

3) histone acetylation and methylation 

4) alterations in chromatin associated proteins: modulators /repressors/ transcription factors... 

5) Microenvironment 

...

Reviewer 3 Report

Very interesting synthesis on the cancer topic. Thanks for that

LBR abréviation must be defined at line 129 and not 164.

There is a problem with the content of lines 585 and 586. Cancers cells are not more sensitive to DNA damaging agents. At the opposite they frequently overexpress DNA repair factor that help them  repair their damages faster and create more genetic instability. This is linked to radio or chemoresistance that are frequently observed after first treatments.

Moreover cancer cells are also more prone to chemo and radio resistance because of an enhancement of their splicing profiles. They create a modified protein repertoire that allow them a better adaptability to their environment and the treatments. This aspect may be linked to a modification of the nuclear speckles architecture in cancer cells. This aspect should be treated.

Round 2

Reviewer 1 Report

Authors' response and revised manuscript is satisfactory.